# Antioxidant Activity and Anti-Nutritional Factors of Selected Wild Edible Plants Collected from Northeastern Ethiopia

**DOI:** 10.3390/foods11152291

**Published:** 2022-08-01

**Authors:** Endale Adamu, Zemede Asfaw, Sebsebe Demissew, Kaleab Baye

**Affiliations:** 1Department of Plant Biology and Biodiversity Management, College of Natural Sciences, Addis Ababa University, Addis Ababa P.O. Box 3434, Ethiopia; zemede.asfaw@aau.edu.et (Z.A.); sebsebe.demissew@aau.edu.et (S.D.); 2Department of Biology, College of Natural & Computational Sciences, Debre Tabor University, Debre Tabor P.O. Box 272, Ethiopia; 3Center for Food Science & Nutrition, College of Natural Sciences, Addis Ababa University, Addis Ababa P.O. Box 3434, Ethiopia; kaleab.baye@aau.edu.et

**Keywords:** DPPH, flavonoid, FRAP assay, oxalate, phenol, phytate, tannin

## Abstract

In Ethiopia, wild edible plants (WEPs) offer a natural food supply for humans to alleviate food insecurity and hunger. Despite the extensive usage of WEPs in Ethiopia, there have been few investigations on their nutritional composition. Our study aimed to evaluate the antioxidant activity and anti-nutritional factors of the most commonly consumed WEPs in Northeastern Ethiopia. The antioxidant parameters including total phenol, total flavonoid, 2,2-diphenyl-1-picrylhydrazyl (DPPH) and Ferric Antioxidant Power (FRAP) assay contents and the anti-nutritional parameters including oxalate, phytate and tannin contents of the selected seven WEPs were evaluated using standard food analysis techniques. The total phenol (mg GAE/100 g) and total flavonoid (mg QE/100 g) content of WEPs resulted in ranges of 0.79–17.02 and 2.27–7.12, respectively. The antioxidant activity revealed that leaves of *Amaranthus hybridus* and *Rumex nervosus* have the highest DPPH and FRAP value, scavenging 50% of free radicals under 50 µg/mL. Non-food values resulted in the respective ranges of 3.37–11.73 mg/100 g oxalate, 16.31–165 µg/100 g phytate and 1.38–5.49 mg/100 g tannin. The investigation indicates that the antioxidant activity of WEPs under research was higher than common crops, and the non-food values were laid in the safe limit, indicating that these might be used for making more healthy and nutritious foods.

## 1. Introduction

Phenolic compounds, vitamin A and carotenoids in plants have antioxidant properties and can boost protection against various diseases. Studies have also indicated that eating plant-based foods minimizes the risk of dying from chronic diseases [1].

The risk of cancer and heart disease can be reduced by many natural products (especially phenolic compounds) in plants. Thus, the interest in naturally occurring antioxidants to be used in food products has expanded substantially. Consumption of wild edible plants (WEPs) can be effective in reducing an excess supply of inflammation such as in common cancer, which may cause many kinds of chronic diseases [2].

Anti-nutritional factors such as phytic acid, tannin, saponin and oxalic acid have adverse effects on nutrients required by the body that inhibit protein digestion, growth and iron and zinc absorption [3]. Anti-nutrients can affect the nutrients that humans need to absorb [4].

Anti-nutritional factors such as oxalate and tannin were high in edible plants in Ethiopia. Some of the plants that had high levels of these factors were *Ximenia caffra*, *Amaranthus graecizans*, *Portulaca quadrifida* [5] and *Opuntia ficus-indica* [6]. The phytate content was also high in some of the Ethiopian wild edible plants such as *Moringa Stenopetala* [7], *Ensete ventricosum* [8] and *Oxytenanthera abyssinica* [9].

The majority of the WEPs were located in the wild, and little is known about their nutritional benefits [10]. Several anthropogenic activities, such as fuelwood harvesting, agricultural development and overgrazing, have put wild edible plants in jeopardy in their natural settings, and this practice hurts the Ethiopian wild edible plants and results in a decrease in commercially important food plants [11]. Studies on the nutritional contents of Ethiopian WEPs are also limited [12]. Similarly, insufficient attention is given to research on the dietary values and anti-nutritional factors of WEPs in the Lasta District. Therefore, this research aims to evaluate the antioxidant activity and anti-nutritional factors of the most consumed wild edible plants in the Lasta District.

## 2. Materials and Methods

### 2.1. Description of the Study Area

The research was carried out in the Lasta District, North Wollo Zone, Amhara Regional State of Northeastern Ethiopia (Figure 1). Lasta is one of the 166 districts in the Amhara Region, with a total of 23 Kebeles and a total population of 119,482, out of which 60,038 are males and 59,444 are females [13]. The district’s population is largely dependent on mixed agriculture, with an average land tenure of 0.5–0.65 hectares per farmer household. According to the Lasta District Agricultural Office [14], Lasta is one of the region’s food-insecure and drought-prone districts. Lasta is bordered to the north by the Gazigibla District (Waghimera Zone), to the south by the Meket District, to the east by the Gidan District and to the west by the Bugna District [14].

The district is located between 11°45′–12°25′ N and 38°45′–39°20′ E with a total area of 968 km^2^. The area is dominated by uplands and has an elevation range of 1680 to 4286 m above sea level [14].

The average annual rainfall is 770 mm. The mean minimum and maximum study area temperatures are 11.9 °C and 27.5 °C, respectively, while the mean annual temperature is approximately 19 °C [14].

According to the current classification of Ethiopian vegetation [15], there are two forms of vegetation in the area: the Dry Evergreen Afromontane Forest and grassland complex (DAF) and the Afro-Alpine belt (AA). The two vegetation types are characterized by the presence of indicator species. The Abune Yosef Mountain area represents the AA vegetation type and the Yimrehane Kristos Church Forest area is one of the representatives of DAF in the study area. The AA is characterized by the presence of *Lobelia rhynchopetalum*, *Hypericum revolutum*, *Erica arborea*, *Festuca simensis*, etc. The DAF is also described by the presence of characteristic species such as *Juniperus procera*, *COlea europaea* subsp. *cuspidata*, *Carissa spinarum*, *Calpurnia aurea*, *Clausena anisata*, *Clutia abyssinica*, *Grewia ferruginea*, *Discopodium penninervium*, *Euclea racemosa*, *CMaesa lanceolata* and *Rhus natalensis*.

*Brassica carinata* (Ethiopian mustard), *Eragrostis tef* (teff), *Hordeum vulgare* (barley), *Lycopersicon esculentum* (tomato), *Solanum tuberosum* (potato), *Triticum aestivum* (wheat), *Vicia faba* (fava bean) and *Zea mays* (corn) are the common crops in the Lasta District [14]. The Agriculture and Rural Development Office of Lasta District gave us written authorization to collect field data and conduct our research.

### 2.2. Plant Sample Collection for Determination of Antioxidant Activity and Anti-Nutritional Factors

The top seven WEPs identified in the study area [16] with their edible parts (grains and leaves of *Amaranthus hybridus*, young shoots of *Rumex nervosus* and leaves of the other five species such as *Erucastrum arabicum*, *Erucastrum abyssinicum*, *Haplocarpha rueppelii*, *Haplocarpha scimperii* and *Urtica simensis*) were collected from different localities of the Lasta District in the morning hours of October 2020 where the edible parts are diversely available. The edible parts of each species were collected from three and more individuals, and the composite sample was taken for analysis. The collected samples were cleaned with distilled water, dried under the shade and stored in the Debre Tabor University Biology Laboratory under freezing conditions. The powdered samples were stored in dark brown umber bottles before being sent to the Addis Ababa University Food Science and Nutrition Center for antioxidant activity and anti-nutritional factors analysis.

### 2.3. Determination of Antioxidant Activities

The total phenolic content of the dried methanol extract generated from the freeze-dried sample was determined using the Folin-Ciocalteu reagent [17]. One mL of extract (1 mg/mL) or standard gallic acid (20, 40, 80, 120, 160, 200 μg/mL) methanolic solution was applied to 2.5 mL of 10% Folin-Ciocalteu reagent dissolved in distilled water and incubated for 5 min. The absorbance was measured with a UV-VIS spectrophotometer (Lambda35, Perkin Elmer, Waltham, MA, USA) at 765 nm after adding 2 mL of 7.5% Na_2_CO_3_ solution to the mixture and incubating it in the dark for 30 min at 24 °C. A blank was made out of 1 mL of methanol, 2.5 mL of 10% Folin-Ciocalteu reagent and 2 mL of 7.5% Na_2_CO_3_ solution at the same time.

The total flavonoid content was also spectrophotometrically measured at 415 nm [17]. One mL of methanolic solution extract or standard solution of quercetin (20, 40, 80, 120, 160, 200 μg/mL) was treated with 0.5 mL of 5% NaNO_2_ solution and 0.5 mL of 10% AlCl_3_ solution. Two mL of 4% NaOH solution was applied and incubated at room temperature for 15 min, and the absorbance was measured against the blank at 415 nm using a UV-VIS spectrophotometer (Lambda35, Perkin Elmer, USA). Without adding a sample or standard, the blank reagent was prepared by adding the whole reagent. Using standard quercetin, a calibration curve was constructed.

The antioxidant activity of WEP samples was carried out with stable free radical DPPH [18]. For each extract, 1 mL of methanolic solution (20, 40, 80, 120, 160 and 200 μg/mL) was applied to 3 mL of DPPH working solution (0.1 mM DPPH in methanol) and incubated for 30 min in the dark. A UV-VIS spectrophotometer (Lambda35, Perkin Elmer, USA) was used to measure the absorbance of extracts at 517 nm, and the results were compared to that of standard ascorbic acid at similar doses. One mL of methanol with a 3 mL working DPPH solution acted as a blank solution.

The [19] approach was used to conduct the FRAP test. The FRAP reagent was made at a volume ratio of 10:1:1 by combining 300 mM sodium acetate buffer (pH 3.6), 10 mM TPTZ dissolved in 40 mM HCl and 20 mM ferric chloride. Methanol was used to dissolve various quantities of the sample (400–800 mg/mL). Of each sample concentration, 25 μL was applied to 175 μL of the FRAP reagent. The blank test included 25 μL FRAP reagent and 175 μL sodium acetate buffer. The absorbance of the mixtures was measured using a UV-spectrophotometer (Lambda35, Perkin Elmer, USA) at 593 nm. The experiment was carried out in triplicate, and ascorbic acid was employed as a positive control in methanol.

### 2.4. Determination of Anti-Nutritional Factors

The oxalate content of WEPs was calculated using the [20] method. The following 3 steps were included within the procedure:

**Step 1—Digestion:** 1 g of sample was on hold in 190 mL of distilled water held in a 250 mL volumetric flask; 10 mL of 6 M HCl was added and digested for 1 h at 90 °C, accompanied by cooling; and then up to 250 mL of distilled water was added before filtration.

**Step 2—Precipitation of oxalate:** a beaker was filled with 125 mL of the filtrate and 4 drops of methyl red indicator were applied. Then 1 mL of additional 6 M HCl was applied, followed by the addition of a concentrated solution of NH_4_OH (used for persistence of yellow color) (dropwise) until the test solution shifted from its salmon pink color to a slight yellow color (pH 4–4.5), heated to 90 °C, cooled and filtered to extract the ferrous ion-containing precipitate. Then, 10 mL of 5% CaCl_2_ solution was added to the filtrate while continuously stirring. It was refrigerated after heating and left at 5 °C overnight. The solution was then centrifuged for 5 min at a speed of 2500 rpm. The supernatant was decanted and fully dissolved in 10 mL of 20 percent (*v*/*v*) H_2_SO_4_ solution.

**Step 3—Titration of permanganate:** up to 300 mL of the total filtrate resulting from the digestion of 1 g of flour was prepared, 125 mL of filtrate was heated until near-boiling, titrated to 0.5 M standardized KMnO_4_ and a solution of slight pink color that lasted for 30 s.

The phytate content over the processed samples was calculated using the [21] methods. About 0.1 g of dried WEP parts were extracted at ambient temperature for 1 h with 10 mL of 2.4% HCl in a mechanical shaker and centrifuged at 3000 rpm for 30 min. The clear supernatant was aged for estimating the phytate content. In 3 mL of the sample solution, a 1 mL Wade reagent (containing 0.03% FeCl_3_·6H_2_O solution and 0.3% sulfosalicylic acid in water) was utilized, and the mixture was combined with a vortex mixer for 5 s. Using UV-Vis spectrophotometer (Lambda35, Perkin Elmer, USA), the absorbance of samples was measured at 500 nm. A serious standard solution containing 5, 10, 15 and 20 μg/mL of phytic acid (sodium phytate analytical grade) was prepared along with 2.4% HCl. In 15 mL, concerning centrifuge tubes including 3 mL of water, a 3 mL standard was added as a blank. In each test tube, 1 mL of the Wade reagent was added, and the solution was mixed with a vortex mixer for 5 s. The mixture was once centrifuged for 10 min, and the solution absorbance was measured by calibrating the spectrophotometer at 500 nm.

The samples’ condensed tannin content material was determined using the [22] method. In the screw cap test tube, 1 g of the sample was weighted. At room temperature with a mechanical shaker, the sample solutions were extracted with 10 mL of 1% HCl in methanol for 24 h. The solution was centrifuged at 100 rpm for 5 min, 1 mL supernatant was taken and a 5 mL Vanillin-HCl test was added. Forty mg of D-catechin was once weighed, and 1% HCl was dissolved into 1000 mL over methanol, which was used as the stock solution. The test tubes were adjusted with 0.0, 0.2, 0.4, 0.6, 0.8 and 1 mL of stock solution, and the volume of each test tube was changed to 1 mL with 1 percent HCl in methanol. Each test tube was supplemented with 5 mL of vanillin-HCl reagent. Then, the UV-Vis spectrophotometer measured the absorption regarding the options or the solution at 500 nm.

### 2.5. Statistical Analysis

Data analysis was carried out using SPSS (version 23). One-way ANOVA at a 95% confidence level was conducted to assess the degree of significance between dietary values and anti-nutritional factors, and Duncan’s test was used to check the significance difference of dietary values and anti-nutritional factors among the tested WEPs. All results were carried out in triplicate, and findings were reported as mean ± standard error.

## 3. Results and Discussion

### 3.1. Total Phenolics and Flavonoids

The leaves of *Haplocarpha rueppelii* had the highest phenolic content (17.02 mg GAE/100 g), whereas the leaves of *Urtica simensis* had the lowest (0.79 mg GAE/100 g). The highest flavonoid content was detected in the leaves of *Amaranthus hybridus* (7.12 mg QE/100 g), and the least amount (2.27 mg QE/100 g) was recorded from the young shoots of *Rumex nervosus* (Table 1).

The overall phenolic content of the studied WEPs is lower than that of African Cabbage (*Cleome gynandra*) [23], lower than *Amaranthus caudatus* grain (257 mg GAE/100 g) in Ethiopia [24] and lower than the total phenolic contents of Chinese wild *Passiflora foetida* [25]. The higher phenolic substance in WEPs has been illustrated to be valuable in the avoidance of various chronic diseases [26].

The total flavonoid content of WEPs collected from the Lasta District falls between 2.27 mg QE/100 g to 7.12 mg QE/100 g (in the young shoots of *Rumex nervosus* and in the leaves of *Amaranthus hybridus* respectively). This variability makes the WEPs a good candidate for the exploration of antioxidants. The result is comparable with WEPs in South Africa [27] and higher than the total flavonoid contents of *Amaranthus caudatus* grain (0.68 mg QE/100 g) [24]. It has been found that flavonoids have antioxidant and free radical scavenging activities and may also protect membrane lipids from oxidation [28].

### 3.2. Radical Scavenging Activity Using DPPH

The DPPH free radical scavenging activity of the extracts of the studied WEPs was estimated by comparing with an ascorbic acid standard at different concentrations (20, 40, 80, 120, 160 and 200 µg/mL) (Table 2). Extract of *Rumex nervosus* young shoots at 200 µg/mL presented a greater percent of inhibition (97.30%) as compared to extracts of other WEPs, and it is very close to the percent of inhibition of the ascorbic acid standard (97.68%). The least percent of inhibition was recorded from the seeds of *Amaranthus hybridus* (52.31%) at a concentration of 200 µg/mL.

Antioxidant molecules can reduce the free radicals of DPPH by contributing electrons and forming a colorless stable molecule 2, 2-diphenyl-1-hydrazine, which decreases the absorption of the solution at 517 nm [29].

The IC50 values of the WEPs extracts were calculated to determine the number of extracts needed to quench 50% of radicals, and the result is presented in Figure 2. At the lower concentrations, the scavenging potential of *Amaranthus hybridus* leaf extract is better than the ascorbic standard and has a good potential to scavenge 50% of free radicals (laid at the concentration of 24.91 µg/mL) that is relatively close to the IC50 value of the ascorbic acid standard (14.74 µg/mL), and the least IC50 value was recorded in the seed extracts of *Amaranthus hybridus* (197.22 µg/mL).

The result is lower than the findings of [29] on the promising wild edible fruits in India and larger than the findings of [24] on the different colored *Amaranthus caudatus* seeds in Ethiopia.

### 3.3. Ferric Reducing Antioxidant Power (FRAP) Assay

The antioxidant activity of the studied WEPs using FRAP was estimated by using FeSO_4_ standard and comparing it with ascorbic acid at different concentrations (50, 175 and 300 µg/mL).

The ferric ion reducing activity power assay of *Haplocarpha schimperi* leaf extract increased, due to the formation of the Fe^2+^-TPTZ complex with increasing concentration, and showed a higher FRAP assay value (529.82 mM) followed by *Amaranthus hybridus* leaf extract (485.45 mM) at 300 µg/mL. The extract of *Amaranthus hybridus* seeds with a concentration of 300 µg/mL had the lowest FRAP assay value (139.25 mM). All studied WEP extracts had better FRAP assay values than the control ascorbic acid and FeSO_4_ standard, except for the extract of *Amaranthus hybridus* seeds (Table 3). The FRAP test assessed the phenolic ability to reduce Fe (3+) to Fe (2+) [29].

The IC50 value of WEP extracts in the FRAP assay was calculated to determine the extent of extracts needed to reduce 50% of ferric ions, and the result is presented in Figure 3. At the lower concentrations, the reducing potential of *A. hybridus* leaf extract is the best to reduce 50% of the ferric ion with low concentration (8.22 µg/mL), followed by *H. schimperi* leaf extract (27.32 µg/mL). The least IC50 value was recorded in the *R. nervosus* young shoot extracts (71.48 µg/mL).

The result is greater than the findings of [29] on the promising wild edible fruits in India. Because FRAP values are dependent on ferric ion reduction [30], higher FRAP values in the tested WEPs entail higher antioxidant activity.

### 3.4. Anti-Nutritional Factors of WEPs

The oxalate, phytate and tannin content of selected WEPs in the Lasta District is low (Table 4). The oxalate content is highest in the leaves of *Erucastrum abyssinicum* (11.73 mg/100 g) followed by in the leaves of *Haplocarpha rueppelii* (9.09 mg/100 g) and in the leaves of *Haplocarpha schimperi* (5.57 mg/100 g). The seeds of *Amaranthus hybridus* have the lowest oxalate level (3.37 mg/100 g). The oxalate content of the studied WEPs is lower than the oxalate contents of raw *Colocasia esculenta* (243.06 mg/100 g) grown in Ethiopia [31], lower than the oxalate content of *Manihot esculenta* (27.16 mg/100 g) grown in Ethiopia [32] and larger than the oxalate content of *Kedrostis africana* (0.28%), a WEP in South Africa [33]. Food containing oxalate promotes tongue discomfort, reduces calcium absorption and increases the production of kidney stones [34].

The phytate content of WEPs in the study area falls on 16.31 µg/100 g (in the young shoots of *Rumex nervosus*) and 165.95 µg/100 g (in the leaves of *Erucastrum arabicum*). The phytate content of the studied WEPs in the study area is comparable with the phytate content of raw *Colocasia esculenta* (115.43 mg/100 g) grown in Ethiopia [31], lower than the phytate content of *Manihot esculenta* (803.95 mg/100 g) grown in Ethiopia [32] and larger than the phytate content of *Kedrostis africana* (2.4%), a WEP in South Africa [33]. The phytate content is highest (165.95 µg/100 g) in the leaves of *Erucastrum arabicum* followed by in the leaves of *Haplocarpha schimperi* (165.92 µg/100 g), in the seeds of *Amaranthus hybridus* (165.88 µg/100 g) and the leaves of *Haplocarpha rueppelii* (165.86 µg/100 g). Because it interferes with the human body’s daily functions including digestion and protein breakdown, phytic acid is referred to as an anti-nutrient [35]. This traps essential nutrients, limiting their availability in the human body [36]. Nutritional disorders such as rickets and osteomalacia are associated with excessive consumption of phytate-rich diets. This anti-nutrient, however, can be quickly eliminated by soaking, boiling, or frying [37].

The highest tannin content is found in the leaves of *Erucastrum arabicum* (5.49 mg/100 g), followed by in the young shoots of *Rumex nervosus* (9.09 mg/100 g) and in the leaves of *Haplocarpha schimperi* (3.60 mg/100 g). The least tannin content (1.38 mg/100 g) is determined from the leaves of *Haplocarpha rueppelii*. Tannins have long been known to reduce protein, mineral and especially iron bioavailability. Tannin-rich foods are recognized to have low nutritional value because they precipitate proteins, inhibit digestive enzymes and iron absorption and affect the absorption of vitamins and minerals from food [3]. The tannin content of the WEPs tested is lower than that of *Colocasia esculenta* (243.06 mg/100 g) cultivated in Ethiopia [31], the tannin content of 20 plant species in India [38] and the tannin content of triticale (285.56 mg/100 g) reported from Amhara Region, Ethiopia [39].

However, the concentrations of oxalate, phytate and tannin found in this study were below the recommended level and may not be hazardous when consumed.

## 4. Conclusions

The plant materials studied showed strong antioxidant and/or free radical scavenging properties. Bioactive substances, particularly total polyphenols and total flavonoids, are higher in this grain than in wheat, rice and oats. The considerable amount of these bioactive compounds in the studied WEPs, predominately known for their antioxidant action, could have also a role in lowering blood pressure. The inherent anti-nutrients (oxalate, phytate and tannin) in the tested WEPs are within tolerable limits, and it could be also reduced from the current amount by processing especially by boiling and roasting the edible parts.

## Figures and Tables

**Figure 1 foods-11-02291-f001:**
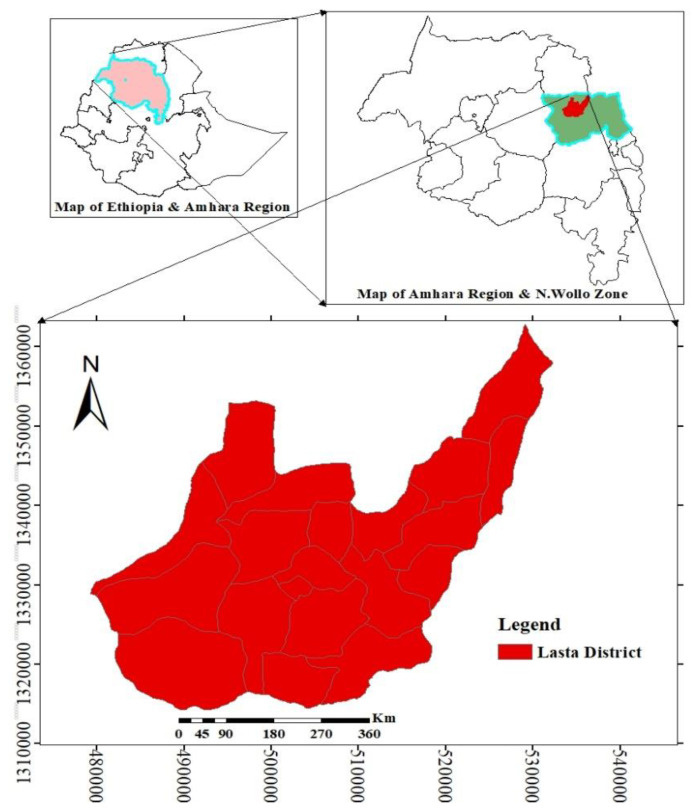
Map of Ethiopia showing the location of the study area (Lasta District) in the Amhara Region.

**Figure 2 foods-11-02291-f002:**
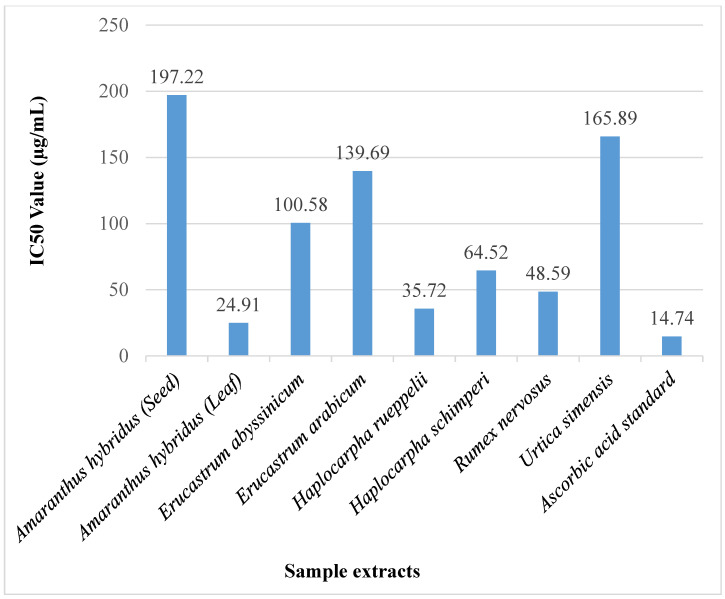
IC50 values in DPPH radical scavenging activity of WEPs.

**Figure 3 foods-11-02291-f003:**
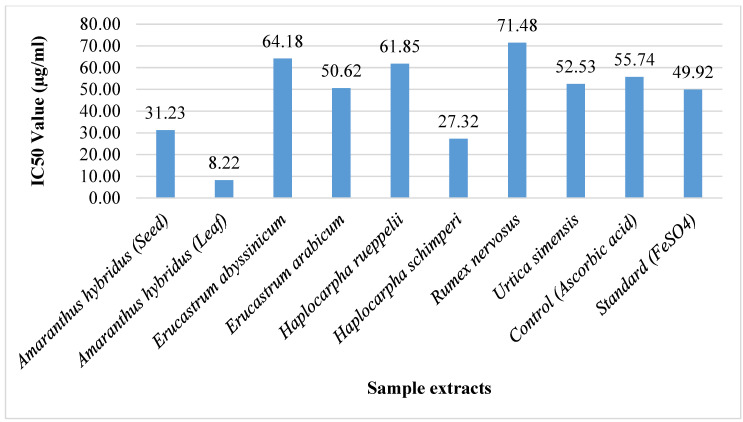
IC50 values in FRAP assay of WEPs.

**Table 1 foods-11-02291-t001:** Total phenolic and total flavonoid contents of selected wild edible plants.

No.	Species	Plant Part	Total Phenols (mg GAE/100 g)	Total Flavonoids (mg QE/100 g)
1	*Amaranthus hybridus*	Grain	10.00 ± 0.01 ^c^	5.97 ± 0.43 ^cd^
		Leaf	13.13 ± 0.05 ^f^	7.12 ± 0.10 ^d^
2	*Erucastrum abyssinicum*	Leaf	12.69 ± 0.00 ^e^	6.23 ± 0.16 ^cd^
3	*Erucastrum arabicum*	Leaf	8.62 ± 0.02 ^b^	3.57 ± 0.29 ^b^
4	*Haplocarpha rueppelii*	Leaf	17.02 ± 0.03 ^h^	5.43 ± 0.12 ^c^
5	*Haplocarpha schimperi*	Leaf	11.69 ± 0.03 ^d^	5.74 ± 0.14 ^c^
6	*Rumex nervosus*	Young shoot	16.28 ± 0.03 ^g^	2.27 ± 0.16 ^a^
7	*Urtica simensis*	Leaf	0.79 ± 0.02 ^a^	5.02 ± 0.40 ^c^

**NB:** The values are the means of three independent composite sample analyses (on a DW basis) ± SE. At *p* < 0.05, different superscripts down the column are significantly different.

**Table 2 foods-11-02291-t002:** DPPH free radical scavenging activities of selected wild edible plants.

Species	Concentration (µg/mL)	% of Inhibition
*Amaranthus hybridus* (Grain)	20	2.74 ± 0.10
	40	12.92 ± 0.24
	80	14.25 ± 0.15
	120	24.63 ± 0.12
	160	33.59 ± 0.22
	200	52.31 ± 0.34
*Amaranthus hybridus* (Leaf)	20	29.03 ± 0.16
	40	55.82 ± 0.34
	80	89.00 ± 0.26
	120	89.11 ± 0.12
	160	91.94 ± 0.11
	200	92.89 ± 0.22
*Erucastrum abyssinicum* (Leaf)	20	14.10 ± 0.27
	40	18.76 ± 0.13
	80	36.78 ± 0.25
	120	69.78 ± 0.11
	160	82.97± 0.18
	200	84.99 ± 0.21
*Erucastrum arabicum* (Leaf)	20	1.61 ± 0.23
	40	5.65 ± 0.14
	80	29.10 ± 0.23
	120	36.63 ± 0.42
	160	51.11 ± 0.31
	200	83.02 ± 0.20
*Haplocarpha rueppelii* (Leaf)	20	40.05 ± 0.16
	40	42.27 ± 0.29
	80	70.25 ± 0.14
	120	85.12 ± 0.10
	160	89.00 ± 0.22
	200	92.96 ± 0.13
*Haplocarpha schimperi* (Leaf)	20	25.93 ± 0.21
	40	44.26 ± 0.24
	80	53.05 ± 0.38
	120	83.70 ± 0.33
	160	88.37 ± 0.11
	200	88.69 ± 0.43
*Rumex nervosus* (Young shoot)	20	39.50 ± 0.28
	40	42.30 ± 0.12
	80	69.40 ± 0.36
	120	73.45 ± 0.10
	160	81.19 ± 0.16
	200	97.30 ± 0.13
*Urtica simensis* (Leaf)	20	15.65 ± 0.22
	40	19.06 ± 0.13
	80	29.46 ± 0.43
	120	35.80 ± 0.15
	160	45.40 ± 0.22
	200	62.25 ± 0.17
Ascorbic acid standard	20	25.98 ± 0.11
	40	45.40 ± 0.20
	80	89.22 ± 0.16
	120	96.87 ± 0.13
	160	97.64 ± 0.23
	200	97.68 ± 0.29

**Table 3 foods-11-02291-t003:** FRAP assay of WEPs.

Species	Concentration (µg/mL)	FRAP Assay (mM)
*Amaranthus hybridus* (Grain)	50	65.75 ± 0.02
	175	71.79 ± 0.04
	300	139.25 ± 0.01
*Amaranthus hybridus* (Leaf)	50	172.51 ± 0.11
	175	180.08 ± 0.24
	300	485.45 ± 0.02
*Erucastrum abyssinicum*	50	74.80 ± 0.02
	175	111.27 ± 0.02
	300	410.90 ± 0.04
*Erucastrum arabicum*	50	80.31 ± 0.11
	175	162.82 ± 0.13
	300	432.39 ± 0.04
*Haplocarpha rueppelii*	50	183.71 ± 0.06
	175	312.53 ± 0.11
	300	471.39 ± 0.23
*Haplocarpha schimperi*	50	134.43 ± 0.02
	175	186.43 ± 0.05
	300	529.82 ± 0.08
*Rumex nervosus*	50	169.67 ± 0.12
	175	371.07 ± 0.06
	300	455.93 ± 0.14
*Urtica simensis*	50	88.13 ± 0.22
	175	130.44 ± 0.17
	300	421.79 ± 0.04
Ascorbic acid	50	45.37 ± 0.15
	175	170.76 ± 0.23
	300	304.04 ± 0.22
FeSO_4_ standard	50	53.93 ± 0.16
	175	167.55 ± 0.28
	300	304.29 ± 0.32

**Table 4 foods-11-02291-t004:** Anti-nutritional factors in selected WEPs.

No.	Species	Plant Part	Oxalate (mg/100 g)	Phytate (µg/100 g)	Tannins (mg/100 g)
1	*Amaranthus hybridus*	Grain	3.37 ± 0.25 ^a^	165.88 ± 0.01 ^ef^	1.71 ± 0.10 ^b^
		Leaf	5.13 ± 0.25 ^bc^	82.81 ± 0.00 ^c^	3.35 ± 0.10 ^d^
2	*Erucastrum abyssinicum*	Leaf	11.73 ± 0.25 ^e^	33.16 ± 0.02 ^b^	3.73 ± 0.25 ^e^
3	*Erucastrum arabicum*	Leaf	5.13 ± 0.25 ^bc^	165.95 ± 0.01 ^g^	5.49 ± 0.16 ^f^
4	*Haplocarpha rueppelii*	Leaf	9.09 ± 0.25 ^d^	165.86 ± 0.01 ^e^	1.38 ± 0.12 ^a^
5	*Haplocarpha schimperi*	Leaf	5.57 ± 0.25 ^c^	165.92 ± 0.00 ^fg^	3.60 ± 0.12 ^de^
6	*Rumex nervosus*	Young shoot	4.99 ± 0.25 ^b^	16.31 ± 0.00 ^a^	5.44 ± 0.20 ^f^
7	*Urtica simensis*	Leaf	5.13 ± 0.25 ^bc^	82.99 ± 0.01 ^d^	2.21 ± 0.13 ^c^

**NB**: The values are the means of three independent composite sample analyses (on a DW basis) ± SE. At *p* < 0.05, different superscripts down the column are significantly different.

## Data Availability

The datasets of this research are available from the corresponding author on reasonable request.

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
