# Peer review of "Antioxidant Activity and Anti-Nutritional Factors of Selected Wild Edible Plants Collected from Northeastern Ethiopia"

_foods, 2022, doi:10.3390/foods11152291_

Round 1

Reviewer 1 Report

The manuscript entitled Antioxidant activity and anti-nutritional factors of selected wild edible plants collected from Northeastern Ethiopia is represents interesting results of antioxidant activity of WEP, but the anti-nutritional factor should include more valuable information as which compounds are involved. 

The issues that should be improved:

  • Materials and methods part should be improved as description of some methods is scarce. Especially the section 2.3 where the determination of antioxidant activities are described. The sentence in line 113-114: "one ml of extract and standard gallic acid methanolic solution..." is unclearly written down. Line 115: why the incubation is needed before adding Na2CO3 solution? Line 118: If samples were incubated before adding Na2CO3 solution, why the bank isn't and correct statement is Folin-Ciocalteu reagent. Line 120: what is nornal solution? Line 126: be more specific if "adding the whole reagent". Line 131: how long it was incubated. Line138: 400-800 g/mL is a lot, did you mean mg/mL? Line 139: Clarify " The blank test included a sample of 25ul and sodium acetate buffer of 175ul. 
  • The section 2.4 determination of anti-nutritional factors should be rewritten according to English language checked (yet horoughly in line 158, ambit anger in line 164, manifest in line 166, about pattern solution in line 167, water brash in line 171, 3ml par in line 173, added yet in line 174...), especially the use of English tenses "...used to be centrifuged...", "...used to be taken..." etc. From the expert point of view, the digestion in step 1 at 100°C it doesn't make any sense. The temperature 100°C is theoretical to high for water solution. The citation " 250 ml of distilled water was added until filtration." did you mean "before filtration"?
  • Table 1: the gallic acid equivalents is abbreviated as GAE in the text, but in table is GA Equ, unify this.
  • IC50 for FRAP assay is not clear how it was determined, as IC50 represents the 50% of inhibition, what is the basic concentration that represents the 50%? 
  • Table 2 and 3: add standard deviation to the results
  • line 266 and further: the result of oxalate content is given due to dry material or extract?
  • comment on statement in line 287: during digestion procedure the extract was boiled to 100°C and the anti-nutrition components are still present as were determined in your results. where is the proof that they are eliminated by soaking, boiling or frying?
  • line 292: is the statement "tannin-rich foods ..." true? Wouldn’t you need to determine which tannins are present in your samples for such a claim? Not all of tannins are harmful. According to this study and statements the identification of present tannins should be done.
  • The conclusion: what is the toxic concentration or that are the limitations of present oxalates, phytate and tannins in food.
  • The references are adequate and used a propriate.

Author Response

Dear reviewer, thank you in advance for your detail comments and I have revised the manuscript and answered your questions as to the document I have attached hereunder.

Reviewer 2 Report

attachment

Author Response

Dear reviewer, Thank you in advance for your detailed comments and questions. I have attached the revised version and attached the document hereunder.

Reviewer 3 Report

The revised version of the article (numerous sections with yellow color) has been improved. The article presents the antioxidant activity and phytochemical content of wild edible plants in Ethiopia. The study has a potential interest for the readers as new sources of food or fortified food systems can be drawn from this research. I consider it relevent to the readership of Foods MDPI journal. In general, the study is original in nature  well revised. However, I have indicated with green color some more corrections that should be provided to improve the body of the text.

Based on the attached comments, I suggest a minor revision prior to further consideration for publication.

Author Response

Dear MDPI foods editor, thank you very much for your valuable comments that you gave us that makes our manuscript scientifically sound and strong.

Round 2

Reviewer 1 Report

The authors provided answers to all the referee's comments, thus satisfying the requirements for publishing an article in this journal.

Author Response

Dear MDPI foods editor, Thank you very much for your valuable comments that makes our paper scientifically sound and strong.